# The Skin Microenvironment: A Dynamic Regulator of Hair Follicle Development, Cycling and Disease

**DOI:** 10.3390/biom15091335

**Published:** 2025-09-18

**Authors:** Weiguo Song, Mingli Peng, Qiqi Ma, Xiaoyu Han, Chunyan Gao, Wenqi Zhang, Dongjun Liu

**Affiliations:** State Key Laboratory of Reproductive Regulation and Breeding of Grassland Livestock, School of Life Sciences, Inner Mongolia University, 235 West University Road, Hohhot 010021, China; weiguosong@mail.imu.edu.cn (W.S.); 32408119@mail.imu.edu.cn (M.P.); 32308296@mail.imu.edu.cn (Q.M.); 0141122082@mail.imu.edu.cn (X.H.); 32208063@mail.imu.edu.cn (C.G.); 32308107@mail.imu.edu.cn (W.Z.)

**Keywords:** hair follicle, skin microenvironment, immune privilege, hair follicle miniaturization, metabolic reprogramming, intercellular communication

## Abstract

As essential skin appendages, hair follicles exhibit complex developmental and regenerative processes shaped by the skin microenvironment. Imbalances in skin microenvironmental homeostasis are often accompanied by follicle miniaturization and even hair loss. In studying the mechanisms of hair follicle development, in addition to focusing on the self-regulation of intrinsic signaling within the follicle, it is also crucial to examine the remodeling of the follicular microenvironment triggered by dynamic changes in the skin microenvironment. Herein, we review the individual and combined roles of various cells, tissues, signaling molecules, and metabolic alterations within the skin microenvironment in hair follicle development. Moreover, we summarize the potential applications of the skin microenvironment in treating hair-related diseases, highlight the existing challenges and limitations in the research field, and provide perspectives on future research directions, aiming to elucidate the critical role of the skin microenvironment in regulating hair follicle development.

## 1. Introduction

In biology, the microenvironment refers to the physical, chemical, and biological conditions surrounding an organ, tissue, or cell, including the extracellular matrix, adjacent cells, tissues, and various signaling molecules. The microenvironment provides suitable living conditions for multiple types of cells and directly or indirectly regulates vital activities such as cell proliferation, differentiation, and apoptosis through secreted factors, thereby serving as a crucial support for organ development and functional maintenance. Early research primarily focused on tumors. With the advancement of human understanding, the roles of microenvironments in other cell types or organs have been further uncovered. Among these, the hair follicle has emerged as a new and promising area of research [1,2,3,4].

Hair follicles are multifunctional skin appendages responsible for hair growth, sensory perception, sebum secretion, and thermoregulation. As a miniature organ, the hair follicle structure is complex, comprising the hair shaft, inner root sheath (IRS), outer root sheath (ORS), and connective tissue sheath [5]. Throughout the mammalian lifespan, hair follicles undergo cyclic phases of growth (anagen), regression (catagen), and rest (telogen) [6]. During different phases of the cycle, the various structures of the hair follicle cooperate closely to ensure normal hair follicle function. Early studies focused on the self-regulation of the hair follicles’ intrinsic signals, often overlooking the role of the surrounding skin microenvironment [7,8]. As the body’s first defense line, the skin protects internal organs, including hair follicles, from external physical, chemical, and biological factors. The skin comprises three layers: epidermis, dermis, and subcutaneous tissue. Within the dermis, hair follicles and abundant blood vessels, nerves, sweat glands, sebaceous glands, and subcutaneous fat comprise the complex hair follicle microenvironment [9]. Various cells and tissues within the microenvironment perform their respective functions in response to external stimuli (temperature fluctuations, epidermal abrasions, mechanical stress, etc.), regulating the adaptation of hair follicle-associated cells to environmental change [10,11,12,13,14,15,16]. However, the mechanisms by which signals from the microenvironment are integrated and regulate hair follicle function have been unclear. Here, we review how cell types, signaling molecules, and metabolic alterations in the skin microenvironment regulate hair follicle cells to perform correct biological functions. We also discuss emerging technologies in hair follicle therapies and further emphasize the roles of the skin microenvironment in hair follicle development, which is crucial for understanding hair-related diseases, developing new therapeutic strategies, and maintaining skin health.

## 2. Hair Follicle Morphogenesis and Cycling

Hair follicle morphogenesis refers to the gradual formation of hair follicles through epidermal and dermal interactions during embryonic development, comprising induction, organogenesis, and cell differentiation (Figure 1). During induction, subepidermal mesenchymal cells aggregate and specific fibroblasts interact with the epidermal placode to form dermal condensates in regularly spaced epithelial areas. During organogenesis, these dermal condensates send “second dermal signals” to the hair substrate, causing it to proliferate and grow toward the dermis to form primary hair follicle germs (HFGs). The HFGs grow downward to form hair pegs, which proliferate and wrap around the dermal condensates to form hair papillae, completing hair follicle development. During differentiation, the hair peg differentiates into the ORS, IRS, and hair matrix cells (HMCs), influenced by the hair papilla. Subsequently, HMCs differentiate into hair shafts that protrude from the surface [17,18,19,20].

A distinctive feature of mature hair follicles is their cyclical nature, which comprises three main phases: anagen, catagen, and telogen [21]. During the anagen phase, the hair follicle is in a state of rapid proliferation, with the hair bulb end flattened, the IRS gradually increasing, and the hair bulb extending into the dermis. Meanwhile, the adipose tissue and vascular system are synchronously remodeled [22]. In the catagen phase, hair follicle growth slows down, the hair papilla begins to atrophy, and the hair root shifts upwards. The bulb end becomes pointed, the cells loosen, the IRS lightens in color, and certain parts break. Hair follicle degradation is primarily caused by programmed keratinocyte death [23]. Finally, the hair follicle remains quiescent during the telogen phase, with reduced HMCs and hair follicle stem cells (HFSCs) activity. These cells stop proliferation and differentiation, awaiting the next growth cycle [24,25].

Hair follicle morphogenesis is divided into three phases. During the induction phase, fibroblasts interact with the epidermis to induce the aggregation of mesenchymal cells, forming dermal condensates beneath the hair placode, which subsequently develop into the hair follicle. During the organogenesis phase, the dermal condensates continue to grow toward the dermis, forming primary hair germ, which later develops into hair pegs. Keratinocytes begin to invaginate and envelop the dermal papilla cells. During the differentiation phase, the hair follicle begins to exhibit distinct structural stratification, with cells possessing differentiation potential forming the inner root sheath, outer root sheath, and hair shaft, ultimately forming a mature hair follicle.

## 3. Key Factors Affecting the Hair Follicle Microenvironment

Hair follicle growth and cyclic turnover are regulated by the skin microenvironment, which comprises diverse cell types and signaling molecules secreted by tissues and cells. These complex components provide essential nutrients and signaling support to hair follicles. Glycolysis, the tricarboxylic acid cycle (TCA), glutamine metabolism, and lipid metabolism are activated to meet energy needs and supply intermediates for rapid hair follicle cell proliferation while adapting to environmental changes [26,27].

Hair follicle morphogenesis and cycling rely on maintaining homeostasis in the skin microenvironment. An imbalance in homeostasis can result in cycle stagnation or hair loss. With age, the skin microenvironment undergoes changes, including a reduction in the number and activity of immune cells, rupture of the arrector pili muscle, mislocalization of sensory nerves, expansion of the basement membrane, and sclerosis of the extracellular matrix. These changes accelerate hair follicle senescence and loss through intercellular communication, signaling, nutrient supply regulation, immunomodulation, and mechanical pressure [28,29,30]. Interestingly, when HFSCs from young mice were transplanted into the skin of aging mice, hair follicles could not develop normally; by contrast, new hair follicles formed when HFSCs from aging mice were transplanted into the skin of young mice [31]. These results suggest that the skin microenvironment is the key determinant of HFSC behavior in terms of self-renewal and differentiation. Therefore, in this section, we will delve into the key factors affecting the hair follicle microenvironment, which include, but are not limited to, various types of cells and tissues, signaling molecules, and altered metabolic levels (Figure 2).

The skin microenvironment is the central support for hair follicle development and cyclic renewal. It involves multiple cell types (e.g., immune cells, fibroblasts, adipocytes, and vascular endothelial cells) and signaling molecules (e.g., hormones, neurotransmitters, cytokines, and reactive oxygen species) secreted by the cells or tissues. Additionally, different hair follicle cell subsets exhibit distinct metabolic profiles, indicating that metabolic pathway reprogramming is an important mechanism by which hair follicles respond to environmental change.

### 3.1. Influence of Cell Types in the Skin Microenvironment on Hair Follicle Development

In the skin microenvironment, there are multiple cell types, among which immune cells, adipocytes, vascular cells, and fibroblasts influence the function of follicle-associated cells directly through cellular interactions or indirectly via secreted factors. Early studies primarily focused on the molecular and cellular establishment of the hair follicle itself, while how different cell types in the skin microenvironment regulate hair follicle development was largely unexplored. With the development of biology, differential expression of genes and proteins due to microenvironmental heterogeneity has garnered attention, and the synergistic effect of various cell types on the hair follicle is now being actively investigated.

#### 3.1.1. Immune Cells

The immune microenvironment of the skin responds continuously to pathogens, making it highly adaptive. It effectively defends against invading pathogens while supporting the normal growth and development of various accessory organs [32]. Understanding the establishment and maintenance of immune cell networks within the complex immune environment of the skin is critical to elucidating the mechanisms of hair follicle regeneration. Similarly to the central nervous system, testes, placenta, and eyes, hair follicles possess a certain degree of immune privilege. Disruption of this status can impair follicular development or cause follicular necrosis [33,34]. Alopecia areata (AA) is an organ-specific autoimmune disease thought to result from immune privilege breakdown [35]. AA pathogenesis is associated with infiltrating immune cells, including T cells and natural killer cells, which erroneously attack hair follicles, leading to hair loss [36]. Clarifying the immune cell-related mechanisms underlying AA pathogenesis is essential for developing effective treatments.

During the hair follicle cycle, the entire skin tissue undergoes extensive remodeling with significant fluctuations in immune cell numbers and activity. In general, the abundances of perifollicular CD4^+^ and CD8^+^ T cells and dermal γδT cells reach a nadir during the telogen phase and peak during the anagen phase, while macrophages progressively decline from mid-telogen to the onset of anagen [37,38]. Although these synergistic changes are well-documented, it remains unclear whether the hair follicle cycle-related changes are a consequence of the cycle or a preparatory step for immune cells to exert primary regulation [11,39]. Recently, the roles of immune cells in hair follicle development have been investigated using specific models involving ablation and hair removal to induce hair follicle cycle synchronization. For example, specific ablation of regulatory T cells (Tregs) during the anagen phase in mice delayed hair follicle development compared with that in the wild-type group. Additionally, early ablation had a more significant effect on hair follicle regeneration than late ablation [40]. Clodronate-induced ablation in early telogen skin macrophages leads to Wnt ligand release, activating HFSCs and accelerating their transition to the anagen phase [11].

Stress-induced alopecia is closely associated with immune cells. In a mouse model, stress was shown to significantly increase the abundance of pro-inflammatory M1-type macrophages in the skin. These macrophages release IL-18 and IL-1β, inducing HFSC apoptosis and leading to hair loss. In contrast, macrophage clearance reduces stress-induced alopecia [41]. Similarly, inhibiting macrophage phagocytosis in skin wound models shifted the wound healing process from fibrosis to regenerative repair, characterized by decreased collagen deposition, attenuated wound contraction, transient Wnt signaling activation, and hair follicle regeneration in the wound area [42]. This functional switch may depend on their spatial localization and polarization state. However, the signaling mechanisms governing macrophage polarization in the microenvironment surrounding hair follicles remain unclear.

Recent studies have revealed new roles for macrophages in HFSC activation and growth cycle regulation. For example, specific subpopulations of macrophages secrete oncostatin M, a cytokine that maintains the resting state of HFSCs and inhibits hair follicle growth [10]. In contrast, Tregs interact with the glucocorticoid receptor to produce transforming growth factor β3 (TGF-β3), which positively regulates hair follicle regeneration following injury by directly activating Smad2/3 signaling in HFSCs, driving telogen to anagen transition [43]. This highlights the cross-communication between Tregs and HFSCs.

Lymphatic vessels, which locally coordinate immune responses and transport immune cells to regional lymph nodes, are distributed in the anterior permanent region of the hair follicle, beginning at the developmental stage and persisting throughout all stages of the cycle, connecting neighboring hair follicles. During the initial stages of physiological stem cell activation or drug-induced hair follicle growth, lymphatic vessels transiently dilate, enhancing tissue drainage and affecting hair follicle growth [44]. Lymphatic capillaries also promote immune cell entry into the hair follicle epithelium, influencing hair follicle cell distribution and differentiation [45,46]. These findings highlight the importance of lymphatics in hair follicle development, cycling, and differentiation, positioning them as integral components of the hair follicle microenvironment. However, the coordinated connectivity and functional impact of the lymphatic vessels during the hair follicle cycle remain poorly understood. Future research should investigate the mechanisms that regulate the interplay between the lymphatic vessels, other immune cells, and hair follicles.

#### 3.1.2. Adipocytes

Adipocytes are located in the dermis and subcutaneous tissues, forming dermal white adipose tissue and subcutaneous white adipose tissue [47]. Adipocytes are highly plastic and capable of metabolic, structural, and phenotypic remodeling in response to various external stimuli. Besides energy storage, adipocytes possess endocrine functions and contribute to the regulation of various physiological processes, including wound healing, bacterial infection defense, and hair growth [48,49]. In the last century, synchronization has been observed between adipose tissue and the hair follicle developmental cycle. The fat layer around the follicle is thickest during the anagen phase and thinnest during the telogen phase [50,51]. This synchronization largely depends on the proportion of immature and mature adipocytes in the tissues adjacent to the hair follicle. During the transition from the telogen phase to anagen phase, precursor cells activate and interact with a larger adipose tissue area. These cells and preadipocytes produce more extracellular vesicles than mature adipocytes, enhancing signal exchange. However, at the end of the anagen phase, restarting the anagen phase is inhibited by reduced signaling and bone morphogenetic protein (BMP) release from mature adipocytes [52,53].

Several studies on mutant mouse models with defects in adipose-associated cells have provided important insights into the interrelationship between adipocytes and the hair follicle cycle. Early B-cell factor 1 knockout mice are born with skin exhibiting adipose-derived stem cell (ADSC) deficiency and reduced dermal adipose tissue. These phenotypes are accompanied by abnormal stagnation of the hair cycle, particularly in the late catagen or telogen phases, and an inability to transition smoothly to the anagen phase [54]. Additionally, the whole-body deletion of peroxisome proliferator-activated receptor γ (PPARγ) in mice is accompanied by loss of adipose precursor cells, leading to poor hair follicle development, disruption of the hair cycle, and exacerbation of perifollicular inflammation. Furthermore, intraperitoneal injection of PPARγ antagonists, such as BADGE or GW9662, depletes preadipocytes and blocks mature adipocyte formation in the mouse skin. This disruption of the adipogenic niche leads to a failure in HFSC activation. As a result, while control mice normally progress into the anagen phase, antagonist-treated mice remain arrested in the telogen phase, manifesting as a functional blockade in hair cycling and regeneration [55,56]. Mice deficient in diacylglycerol acyltransferase (DGAT), a key enzyme in adipocytes, develop hair dryness and alopecia at 6–8 weeks of age. This may be related to DGAT deficiency affecting adipocyte and sebaceous gland development [57].

Culturing ADSCs in a hypoxic microenvironment promotes dermal papillary cell (DPC) growth, potentially related to the upregulation of HIF-1α and ERK1/2 signaling pathways [58]. Recently, scientists have successfully regenerated skin tissue using ADSCs extracted from human adipose tissue and adipose-derived intercellular matrix as the primary bioprinting materials. Additionally, hair follicle-like structures have emerged in the generated skin tissues, presenting a novel approach for constructing human hair follicles in vitro [59]. These findings emphasize the critical role of adipocytes in regulating the hair follicle cycle and reveal the complex interactions between adipocytes and hair follicles.

#### 3.1.3. Vascular Cells

The vascular system in the skin is complex and delicate, essential for maintaining normal skin function. The blood vessels in the skin, primarily found in the dermis and subcutaneous tissue, supply oxygen and nutrients to the surrounding cells and tissues through a network of capillaries that are responsible for removing the metabolic waste produced by the cells [60]. During the anagen phase, the hair bulb in the dermis proliferates rapidly and becomes tightly packed with adipocytes and new capillaries in the subcutaneous tissue. During the catagen phase, the hair bulb undergoes shrinkage, accompanied by the formation of a horizontal vascular plexus from subcutaneous blood vessels beneath the skin surface. This vascular network undergoes remodeling during the initiation of new follicular processes. Notably, the use of angiogenesis inhibitors can delay the entry into the hair follicle anagen phase by interfering with blood vessel formation [61]. Conditioned media from mesenchymal stem cells, containing proteins, lipids, free nucleic acids, and cellular vesicles, have generated widespread interest owing to their potential to induce hair regeneration, likely by promoting angiogenesis [62,63].

Vascular endothelial cells are essential in hair follicle cycle transition. Co-culturing epithelial cells, DPCs, and vascular endothelial cells forms dumbbell-shaped HFGs. Compared with HFGs without vascular endothelial cells, these vascular endothelial cell-containing HFGs showed higher expression levels of hair morphogenesis-related genes in vitro and greater hair shaft regeneration after transplantation in a nude mouse model [64]. Activin receptor-like 1 knockout mice exhibited significant dilation of vascular endothelial cell lumens during the late telogen-to-anagen transition of the hair cycle, accompanied by delayed HFSC activation. This delay is associated with increased secretion of BMP4 signaling molecules by vascular endothelial cells. BMP4 maintains HFSC quiescence and may contribute to delayed HFSC activation when highly expressed in vascular endothelial cells [65]. Furthermore, topical application of quercetin to mouse skin activated HIF-1α, which promoted vascular endothelial cell proliferation and migration, initiating perifollicular angiogenesis and promoting telogen to anagen transition [66].

In addition to the precise synergy of the above cell types in hair follicle development, other cell types, such as fibroblasts, mesenchymal cells, and various stem cells, are also essential (Figure 3). The integration of different cell types within the skin microenvironment and modifications of the hair follicle niche enable the cyclicity of multiple lineages in the follicular structure [67,68,69].

The skin is divided into three layers: epidermis, dermis, and subcutaneous tissue. The hair bulb, at the base of the hair follicle, penetrates deep into the subcutaneous adipose tissue, while the hair shaft penetrates the body surface. Dermal papilla cells in the bulb act as signaling centers and regulate the proliferation and differentiation of hair follicle stem cells within the bulge area. Different cell types (e.g., immune cells, adipocytes, endothelial cells, and fibroblasts) and tissues (e.g., neural, vascular, and lymphatic) surrounding the hair follicle constitute the skin follicle microenvironment, which regulates intercellular communication, signaling, nutrient supply, immune regulation, and mechanical pressure within follicle-associated cells.

### 3.2. Signaling Molecules

Signaling molecules in the skin microenvironment are closely related to key physiological processes during hair follicle morphogenesis and development, including epithelial–mesenchymal transition, immune regulation, and stem cell lineage differentiation. These molecules directly or indirectly regulate hair follicle development by sensing various external stimuli, allowing animals to adjust their hair growth in response to environmental changes. Thus, understanding how different signaling molecules in the skin achieve precise regulation of hair follicles is important for the development of treatments for hair disorders.

#### 3.2.1. Hormones

Hormones are highly effective bioactive substances secreted by endocrine glands or endocrine cells that act as “messengers” in the body to regulate metabolism, respond to external stimuli, and maintain internal homeostasis. They can be categorized into nitrogen-containing hormones and steroid hormones [70,71]. Different hormones also have a significant influence on the growth and development of hair follicles. Androgenic alopecia (AGA) is a genetically predisposed androgen-dependent disease with a complex pathogenesis; however, the role of androgens in AGA has been generally recognized. More specifically, testosterone is converted to dihydrotestosterone (DHT) by 5α-reductase and binds to cellular androgen receptors in the hair follicle, interrupting the hair follicle cycle, leading to hair follicle miniaturization and loss. Interestingly, androgen receptors deficient mice become obese, suggesting a possible link between androgen signaling and adipocytes [72]. The effect of estrogen on hair follicle development deserves our full attention. Researchers have observed several interesting phenomena: for example, hair regrowth is more frequent in spayed pigs than in normal females, whereas topical application of estradiol inhibits hair growth in dogs and mice [73,74,75]. These differences may arise from variations in estrogen receptor expression, metabolic conversion, or interaction with local growth factors. Additionally, hair growth is inhibited during lactation and pregnancy [76]. Indeed, estradiol can significantly alter hair follicle growth and cycling by binding to estrogen receptors and affecting aromatase activity. However, the precise signaling context determining these dual effects remains unclear, warranting further investigation.

In the skin, the hypothalamic–pituitary–thyroid and hypothalamic–pituitary–adrenal axes influence various cell processes, including the hair follicle cycle. Thyroid hormones stimulate hair follicle matrix cells, keratinocyte proliferation, and pigmentation, potentially regulating the energy metabolism required for hair follicle development by affecting mitochondrial activity [77,78]. Meanwhile, glucocorticoids, a class of steroid hormones secreted by the adrenal cortex, are used to treat AA because of their immunosuppressive properties. Although glucocorticoid receptors are expressed in multiple skin cell populations (e.g., keratinocytes, dermal papilla cells, immune cells, and skin fibroblasts), the specific cell types interacting with glucocorticoids during hair loss pathogenesis remain to be elucidated [79]. Nevertheless, corticosterone (a major glucocorticoid in rodents) initiates crosstalk between DPCs and HFSCs. Corticosterone also maintains HFSC quiescence by inhibiting GAS6 expression in DPCs, while restoring GAS6 expression can overcome stress-induced inhibition of HFSC activation and hair growth [80]. These findings demonstrate that corticosterone acts as a microenvironmental regulator to inhibit HFSC activity, providing a potential molecular target for developing intervention strategies.

Stress has long been recognized as a factor in hair graying. It has only recently understood that heightened sympathetic activity during acute stress releases norepinephrine, which binds the β2-adrenergic receptor on melanocyte stem cells (McSCs). This process triggers rapid stem cell growth, leading to their differentiation, migration, and, eventually, the irreversible depletion of the McSCs reservoir causing hair graying [81]. Notably, norepinephrine simultaneously stimulates melatonin secretion, creating a complex regulatory network that modulates hair pigmentation [82]. Melatonin, a nitrogen-containing hormone, is a well-recognized circadian hormone that may contribute to oxidative stress, apoptosis inhibition, and cancer progression. In the skin, melatonin receptors are not only expressed by epidermal keratinocytes, fibroblasts, and endothelial cells but also hair follicle-associated cells. Melatonin can promote Wnt/β-catenin signaling in DPCs by mediating Wnt ligand expression in HFSCs [83]. In adult cashmere goats, melatonin promotes secondary hair follicle development by activating the KEAP1–NRF2 signaling pathway and inhibiting the inflammatory transcription factor NF-κB/AP-1 [83,84]. These findings collectively demonstrate that hormones within the skin microenvironment do not function individually but cooperate with various types of cells or signaling molecules to achieve precise and rapid adaptation to the external environment through a hierarchical signaling pathway. Consequently, significant research is required to elucidate the mechanisms by which the macroenvironment regulates hair follicle development by integrating specific hormone signals.

#### 3.2.2. Vitamins

Vitamins are micronutrients necessary for maintaining normal physiological functions and categorized as fat- or water-soluble. In enzymatic reactions, vitamins act as coenzymes or cofactors and scavenge free radicals to support cellular functions [85,86]. They are also essential for hair follicle health and growth. Among the fat-soluble vitamins, vitamin A is crucial in regulating HFSC fates. All-trans retinoic acid, a vitamin A metabolite, acts as a potent upstream regulator that balances the contribution of HFSCs to epidermal repair and hair regeneration. HFSCs cultured with all-trans retinoic acid accurately mimic stem cell behaviors, including the transition from quiescent state to active self-renewal and the progression toward a differentiated fate [87].

Vitamin D has immunomodulatory effects, and evidence suggests that vitamin D and its receptors may play a role in skin homeostasis [88,89]. Keratinocytes, macrophages, and dendritic cells in the skin can synthesize active vitamin D, contributing to the pathogenesis of various skin diseases [90,91]. The vitamin D receptor is widely distributed among all heterogeneous cells in the skin and the major cell populations of the hair follicle. Hence, its aberrant expression leads to hair follicle cycle abnormalities [92]. Considering that AA is typically characterized by an inflammatory infiltrate around the hair follicle, vitamin D may be a potential therapeutic target due to its immunomodulatory properties [93].

Among the water-soluble vitamins, the vitamin B family is particularly important. Deficiencies in riboflavin (vitamin B2), biotin (vitamin B7), folic acid, and vitamin B12 are associated with hair loss [94]. Biotin acts as a cofactor for carboxylases, which catalyze various metabolic reactions important for maintenance of healthy skin and hair, and biotin-deficient newborns suffer from severe dermatitis and alopecia. Vitamin B12 is necessary for DNA synthesis, nerve function, and erythrocyte formation and participates in the methionine cycle, which contributes to the synthesis of melanin, thus promoting hair pigmentation. Consequently, Vitamin B12 deficiency can lead to hair graying and luster loss [95]. While numerous studies report associations between vitamin B family deficiency and hair loss, many human studies remain observational and lack mechanistic depth. Controlled intervention studies and tissue-specific knockout models are needed to establish causal roles and elucidate downstream mechanisms.

Vitamin C, another water-soluble vitamin, protects hair follicles from free radical damage, helping maintain normal hair metabolism and melanin formation. Scurvy caused by vitamin C deficiency is accompanied by “spiral hair” and perifollicular hemorrhage, which results from a decrease in the number of reduced disulfide bonds, leading to a decrease in keratin cross-linking [96]. Considering the functions of vitamins in hair cycling and immune system defense, further molecular studies are necessary to explore how particular vitamins influence hair growth in patients with AA, aided by clinical data to identify correlations between AA and contributing vitamins.

#### 3.2.3. Growth Factors

Growth factors are peptides that regulate cell growth, proliferation, and differentiation by binding to cell surface receptors. They facilitate cell growth and division, accelerate tissue repair, stimulate neoangiogenesis, regulate immune functions, and contribute to nervous system regulation. They facilitate cell growth and division, accelerate tissue repair, stimulate neoangiogenesis, regulate immune functions, and contribute to nervous system regulation [97,98]. Growth factors operate through autocrine and paracrine mechanisms characterized by high specificity and affinity. Platelet-rich plasma contains at least six essential growth factors, including fibroblast growth factor (FGF), platelet-derived growth factor, vascular endothelial growth factor (VEGF), epidermal growth factor (EGF), TGF-β and insulin-like growth factor-1. VEGF is a key angiogenic factor crucial for angiogenesis [99]. Exogenous VEGF promotes human DPC proliferation in a dose-dependent manner, which can be inhibited by a neutralizing antibody against VEGFR-2 (the primary VEGF receptor) [100]. VEGF markedly reduces 5α-DHT-induced apoptosis of HFSCs in a concentration-dependent manner. Additionally, the 5α-DHT-induced decrease in Bcl-2/Bax ratio and increase in caspase-3 are reversed by VEGF, while the anti-apoptotic effect of VEGF is hindered by inhibition of PI3K/AKT pathway activation [101]. A study on androgenetic alopecia indicated that oral minoxidil may promote hair regrowth by upregulating VEGF expression. In a prospective trial involving 50 participants, 12-week administration of oral minoxidil (1 mg/day) significantly increased serum VEGF levels compared to the control group (217.88 ± 22.65 pg/mL vs. 142.81 ± 23.14 pg/mL, *p* < 0.0001). This elevation in VEGF was accompanied by improved hair count and diameter, reduced shedding, and was strongly correlated with hair regrowth outcomes. The findings suggest that VEGF upregulation plays a key role in the therapeutic mechanism of minoxidil [102]. EGF is important for hair follicle morphogenesis, mainly involved in epithelial cell growth, proliferation, and differentiation. It promotes the proliferation of interfollicular epithelial cells and fibroblasts, and induces hair follicles to enter the catagen phase [103]. Mice lacking EGFR in the skin exhibit adipocyte defects. Therefore, EGF may also affect hair follicle development by interacting with adipocytes [104]. The regulatory mechanisms among these factors remain to be further clarified.

FGFs are widely expressed in many tissues and play an important role in cell proliferation, differentiation, migration, embryonic development, tissue repair, and angiogenesis. Several members of the FGF family regulate the hair follicle cycle. For example, injecting FGF7 into thymus-free nude mice induces hair growth, while administering FGF7 before cytarabine chemotherapy partially ameliorates chemotherapy-induced hair loss [105]. High concentrations of FGF18 promote DNA synthesis in human DPCs, dermal fibroblasts, epidermal keratinocytes, and umbilical vein endothelial cells. In mice, subcutaneous injection of FGF18 induced hair follicles in the telogen phase to enter the anagen phase [106]. In cashmere goats, FGF5 knockdown promotes the growth of secondary hair follicles, increases fleece production, and improves fleece quality [107]. FGF10 has shown promise in wound healing therapy and hair loss treatment, but its instability and immunogenicity as recombinant heterologous factors may affect their biological activity [108]. To therapeutically exploit FGFs, their exact mechanisms of action have to be elucidated, and their production technology improved.

#### 3.2.4. Other Signaling Molecules

Other molecules, in addition to the specific classes of signaling molecules previously mentioned, also contribute to hair follicle development (Table 1). Cyclosporine A, a short peptide comprising 11 amino acids, is an immunosuppressant that inhibits mouse hair follicles from entering the catagen phase and prolongs the anagen phase by promoting stromal cell proliferation. Additionally, cyclosporine A increases the expression of VEGF, HGF, and nerve growth factor [109]. In 2021, American researchers successfully activated transient receptor potential vanilloid in nerve terminals within mouse skin using chemical genetics. This activation encouraged the cyclic regeneration of telogen hair follicles. The study confirmed that TRPV1 activation causes sensory nerves in the skin to release calcitonin gene-related peptide, which induces macrophage apoptosis. This apoptosis further induces specific types of fibroblasts in the skin to express the hair growth factor osteoblastin, enhancing hair follicle growth. The study contributes to the understanding of the repair mechanism of hair follicle damage caused by epidermal abrasion [110].

When the skin and hair follicles are under stress or at critical developmental nodes, non-cellular factors in the microenvironment can promote intercellular communication. This transformation of cellular functions through various signaling pathways contributes to establishing a microenvironment suitable for the dynamic development of hair follicles. The Wnt, SHH, BMP, JAK–STAT, and other signaling pathways have been found to regulate hair follicle development [111,112]. SHH signaling regulates epithelial growth and hair papilla formation by affecting the proliferation of dermal and epithelial cells. This process is essential for epidermal basal cell formation in hair follicles [22]. BMP signaling inhibits cell proliferation and facilitates the transition of the hair follicles from the anagen to catagen phase [65]. Meanwhile, JAK–STAT signaling is a highly efficient system that regulates HFSC activation. Local inhibition of the JAK-STAT signaling pathway using tofacitinib promotes rapid hair growth in mice and humans with AA [113].

The regulatory roles of various signaling molecules in the skin microenvironment ensure the skin can precisely coordinate the proliferation and differentiation of relevant tissues and organs as needed. However, the diversity of signaling molecules and their crosstalk add to the microenvironment’s complexity. Hence, understanding and searching for primary signaling molecules susceptible to perturbation could be a breakthrough in the hair loss treatment.

**Table 1 biomolecules-15-01335-t001:** Effects of signaling molecules in the skin microenvironment on hair follicle development.

Signaling Molecule	Source	Mechanism	Reference
Interferon-γ	Immune cells	Enhances TGF-β 2 immunoreactivity and mRNA transcription levels	[114]
Tumor necrosis factor	Macrophages	Induces AKT/β-catenin signaling in HFSCs	[115]
Prolactin	Pituitary gland	Affects the proliferation and apoptosis of follicular keratinocytes	[116]
Hepatocyte growth factor	Dermal white adipose tissue	Upregulates WNT6 and WNT10B, inhibits SFRP1, and activates Wnt/β-catenin activity	[117]
Signal peptide CUB-EGF like domain-containing protein 3	Dermal cells	Regulated by Hedgehog signaling, activates TGF-β signaling	[118]
Osteoblastin (SPP1)	Melanocytes	Signals to epithelial stem cells in adjacent hair follicles via CD44 receptors	[119]
Neutrophil elastase	Neutrophils	Specifically degrades collagen XVII (COL17A1) via hydrolysis	[120]
ANGPTL7 ANGPTL4	Skin stem cells	ANGPTL7 promotes lymphatic drainage, resulting in capillary lymphatic vessels tightly surrounding the HFSC niche, helping maintain the stem cell resting state. When stem cells are activated, they shift from expressing ANGPTL7 to expressing ANGPTL4. This shift triggers the transient dissociation of lymphatic drainage, promoting stem cell activation and tissue regeneration	[121]
Platelet-derived growth factor	Platelets	Promotes HFSCs proliferation and self-renewal	[122]
Histone deacetylases 1 Histone deacetylases 2	Papilla cells	During anagen, histone deacetylases 1 and 2 protect DPCs from apoptosis by inhibiting P53 activity and maintaining Wnt activity in DPCs to promote hair follicle growth	[123]
Major histocompatibility complex II and Aire	Thymic epithelial cells	Irreversibly transform the hair follicle pluripotent stem cells’ fates	[124]
Thyrotropin-releasing hormone	Hypothalamus	Stimulates keratinocyte proliferation and promotes hair shaft elongation	[125]
Galanin	Nervous tissues	Reduces stromal keratinocyte proliferation and shortens the growth period	[126]

### 3.3. Metabolic Reprogramming

The coordinated interplay between cellular constituents and signaling molecules within the skin microenvironment, as detailed above, necessitates a downstream mechanistic executor. This role is fulfilled by metabolic reprogramming, wherein dynamic shifts in cellular metabolism directly empower the biological behaviors that govern hair follicle fate. Metabolic reprogramming is the process by which cells systematically alter their metabolic patterns under certain physiological or pathological conditions to adapt to environmental changes while satisfying their growth and differentiation needs [127,128]. Similarly, alterations in microenvironmental metabolic levels can influence hair follicle development. Glycolysis and the TCA cycle are fundamental processes in cellular metabolism. They support normal cellular function and signaling while fulfilling cellular energy needs. Additionally, they exhibit significant potential in regulating hair follicle development. At the beginning of the anagen phase, autophagy in HFSCs enhances the glycolytic process by increasing the expression and activity of lactate dehydrogenase, thereby activating HFSCs and facilitating hair regrowth [129]. Meanwhile, as HFSCs differentiate into ORS precursor cells, a metabolic shift occurs from glycolysis to oxidative phosphorylation and glutamine metabolism. Importantly, this metabolic transition must be inhibited at the end of the anagen phase to allow precursor cells to revert to the stem cell state [130]. Additionally, the hypoxic environment induced by skin microorganisms activates HIF-1α signaling pathway in keratinocytes, promoting glutamine metabolism and IL-1β production and inducing hair follicle regeneration. Inhibitors of glutamine metabolism can reduce the expression of the regenerative signals WNT7B and SHH, as well as the stem cell marker KRT15, in keratinocytes, while inducing the expression of the terminal differentiation marker KRT1 [131]. However, the precise regulatory mechanisms underlying the preference of HFSCs for glycolysis and glutamine metabolism require further elucidation.

Obesity induced by a high-fat diet leads to the generation of excessive reactive oxygen species (ROS) and activation of the IL-1R signaling pathway, resulting in lipid droplet accumulation and NF-κB activation within HFSCs. This process significantly inhibits SHH signaling, driving aberrant differentiation and depletion of HFSCs, ultimately accelerating hair follicle miniaturization and hair loss [132]. Intermittent diets (16/8 time-restricted feeding and alternate-day fasting) inhibited hair regrowth in mice, and this effect was not associated with reduced calorie intake, circadian rhythm changes, or mTORC1 cellular nutrient-sensing mechanism. Intermittent fasting enhanced the communication between the adrenal glands and dermal adipocytes, triggering the release of free fatty acids, which entered the microenvironment of HFSCs to disrupt their normal metabolism and enhance ROS production, leading to apoptosis [133]. Cashmere goat is an important model for studying hair follicle development. Metabolomic studies of skin hair follicle tissues from different stages of the hair follicle development cycle and under different feeding modes (grazing or barn raising) revealed that the expression patterns of metabolites such as sugars, lipids, amino acids, and nucleotides in skin tissues affect hair follicle growth. In particular, 2′-deoxyadenosine, L-valine, 2′-deoxyuridine, riboflavin, cytosine, deoxyguanosine, L-tryptophan, and guanosine-5′-monophosphate may regulate the hair follicle cycle through their involvement in ABC transporter protein [134].

Metabolic reprogramming is crucial for adapting to environmental changes during hair follicle development, defending against external stresses, supporting cell proliferation, meeting biosynthetic requirements, and protecting cells from oxidative stress and apoptosis. However, the mechanisms by which hair follicles sense metabolic shifts and adjust cell fate are unclear. As our understanding of metabolic reprogramming improves, new strategies for metabolite replenishment and stem cell metabolic control may emerge, offering innovative approaches for treating hair loss and regenerating hair follicles.

## 4. Interventions Targeting the Skin Microenvironment to Improve Hair Follicle Growth

Interventions targeting the skin microenvironment to treat hair loss have gained theoretical support from basic research and achieved initial therapeutic success in clinical settings. Minoxidil is an internationally recognized topical treatment for AGA, promoting hair growth by dilating scalp blood vessels, enhancing microcirculation, and stimulating hair follicle epithelial cell proliferation [135]. However, minoxidil may cause side effects and lead to dependence problems. To address these challenges, researchers have proposed a new material biomodulation therapy, which utilizes polydopamine nanoparticles to deliver ROS and promote hair follicles to enter the anagen phase, resulting in hair growth [136]. Nonetheless, ensuring accurate delivery and establishing a safe dosage of administered substances continue to be critical challenges. Microneedling therapy is a relatively novel treatment for AA that uses fine needle-like instruments to create minimally invasive mechanical damage to the skin’s soft tissue, activating the skin’s repair mechanisms and promoting hair follicle growth [137,138]. This process triggers the release of growth factors—including PDGF, EGF, FGF, and TGF. These factors promote angiogenesis, neocollagenesis, and modulation of the hair cycle [139,140]. Alternatively, Liebl et al. propose that microneedling may not produce conventional wounds. Instead, its effects may stem from electrically stimulated cell proliferation: as microneedles approach cell membranes, the Na^+^/K^+^-pump activates, rapidly elevating the electrical membrane potential from approximately −70 mV to −100 mV. This hyperpolarization enhances local cellular activity and triggers the release of proteins, potassium, and growth factors into the extracellular space. Consequently, fibroblasts migrate to the site, facilitating collagen production and creating a regenerative microenvironment conducive to hair follicle activation and growth [141]. Autologous platelet-rich plasma has the potential for hair loss treatment. After collecting venous blood from patients and centrifuging it to concentrate platelets, white blood cells, and other components, the autologous blood with a high concentration of growth factors is created. This concentrated solution is injected into the superficial dermis of the scalp to stimulate the production of collagen, elastic fibers, and blood vessels, thereby promoting hair growth [142]. In recent years, low-energy laser therapy has gradually emerged as a promising treatment, which delivers therapeutic effects through low-intensity lasers and light-emitting diodes in infrared and visible red wavelengths targeting injured areas. Low-intensity lasers refer to wavelengths within the red to near-infrared regions of the spectrum, specifically between 660 and 905 nm. These wavelengths can penetrate the skin and tissues, enhance scalp blood flow, reduce perifollicular inflammation, and increase ATP production to support cell metabolism. Additionally, the upregulation of extracellular matrix proteins can enlarge the size of DPCs, reverse follicle miniaturization, and promote hair growth [143]. Despite their promising potential, these emerging therapeutic strategies are currently in the research phase, and their efficacy and safety require further validation.

## 5. Challenges and Prospects

Although substantial advances have been made in understanding the skin hair follicle microenvironment, significant challenges and limitations persist. Hair follicle development is a complex process reliant on the precise regulation of multiple genes and signaling pathways. Disruption of this process can lead to impaired follicular development and pathologies such as hair loss. Both intrinsic and extrinsic signaling pathways coordinate to drive follicular morphogenesis and cycling: intrinsic signals maintain basal cellular states, whereas extrinsic cues modulate development in response to metabolic supply and cellular demands [144,145]. While Wnt/β-catenin activation is known to be essential for anagen initiation, and factors such as BMP and FGF contribute to catagen onset, the precise signals that definitively trigger the beginning and end of the hair cycle remain unclear. The existence of a master “pacemaker” cell population or signaling hub remains controversial. HFSCs demonstrate remarkable plasticity. An important unanswered question is how physical factors (e.g., extracellular matrix stiffness), chemical signals (such as cytokines and metabolites), and biological components (including immune cells and nerve endings) integrate within the microenvironment to dictate stem cell fate—whether they remain quiescent, self-renew, differentiate into hair follicle lineages, or adopt an epidermal identity during wound healing. Furthermore, the dynamic establishment and maintenance of immune privilege below the bulge region of anagen follicles are not fully understood. Key uncertainties include the specific immunosuppressive molecules and cell types involved, and how this protection is disrupted under conditions of stress, disease, or aging, potentially leading to autoimmune-mediated hair loss. We also face challenges in translating these findings from technology to clinical application. Current models rely heavily on mouse studies, yet fundamental differences exist between human and murine hair cycling—human follicles exhibit mosaic growth patterns, whereas murine cycles are synchronous. This discrepancy complicates modeling human conditions like AGA in mice, thereby impeding therapeutic development. More physiologically relevant human disease models are urgently needed. Moreover, even if potent growth factors or small molecules are identified, delivering them in an efficient, sustained, and safe manner to specific follicular cells remains a challenge. Controlling their spatial and temporal activity to avoid adverse effects—such as hypertrichosis or carcinogenic risk—is equally critical. Thus, a deeper dissection of the composition and spatial architecture of the skin microenvironment is imperative.

Advances in molecular and cellular biology have shifted the focus of hair follicle research from traditional morphological observation to in-depth investigation of underlying molecular mechanisms, offering new avenues to address long-standing challenges. The rapid development of high-throughput sequencing technologies, particularly the integration of single-cell and spatial transcriptomics, has enabled the construction of high-resolution spatial cell atlases [143,146]. These efforts have helped characterize the transcriptional profiles, functional states, and pathological features of diverse cell types within the skin—including epidermal, follicular, melanocytic, endothelial, and immune cells. Furthermore, lineage-tracing studies have delineated differentiation trajectories between hair follicle stem cells and dermal papilla progenitors, and have linked specific cell subpopulations to molecular signatures of common skin disorders [147,148,149]. These insights provide a deeper understanding of microenvironmental remodeling during follicular development. The emergence of organoid technology has markedly advanced the field. In 2016, the Tsuji team generated a functional epidermal system containing hair follicles and sebaceous glands from mouse iPSCs. Through transplantation of embryoid bodies derived from induced pluripotent stem cells, they reconstituted full-thickness skin and subcutaneous adipose tissue [150]. In 2020, the Koehler laboratory adapted this strategy to human cells. By adding LDN and FGF on day 3 of differentiation, they induced the development of epithelial cysts surrounded by cranial neural crest cells, with visible hair emergence by day 70. After extended culture, the organoids resembled 18-week fetal skin, comprising stratified epidermis, adipocyte-rich dermis, and pigmented hair follicles. Upon implantation into nude mice on day 140, the organoids generated planar skin complete with vasculature and folliculo-sebaceous units. This reproducible system has become a state-of-the-art tool for disease modeling [151]. Interdisciplinary approaches incorporating materials science, systems biology, and computational modeling have further enabled the in vitro reconstruction of dynamic hair follicle developmental processes [152]. The Miao laboratory used extrusion-based bioprinting to fabricate regenerative mouse skin with layered dermal, intermediate, and epidermal structures. Incorporation of DPCs was critical for de novo hair follicle formation, yielding follicular-like structures within 7 days that supported robust hair growth after transplantation [153]. Similarly, Karande’s team successfully integrated human hair follicles into bio-printed skin constructs that recapitulated native tissue architecture—though their functional cycling capacity remains to be validated in vivo [154,155]. Such models are invaluable for predicting how microenvironmental changes influence follicular development.

Among mammalian organs, hair follicles are exceptional in their capacity for lifelong cycles of degeneration and regeneration. This cyclical behavior requires hair follicle stem cells (HFSCs) to repeatedly transition between quiescent, activated, and differentiated states. Well-defined anatomical regions and distinct progenitor populations allow precise fate mapping throughout the cycle. Clinically, aging-related lengthening of the telogen phase is a common phenomenon, making the hair follicle a compelling model for studying systemic aging mechanisms and evaluating anti-aging strategies—such as senescent cell clearance. These attributes establish the hair follicle as a powerful system for probing stem cell plasticity, cellular aging, and evolutionary biology. We anticipate that ongoing research into the skin microenvironment will continue to unveil new mechanisms in follicular biology and adult stem cell function.

## Figures and Tables

**Figure 1 biomolecules-15-01335-f001:**
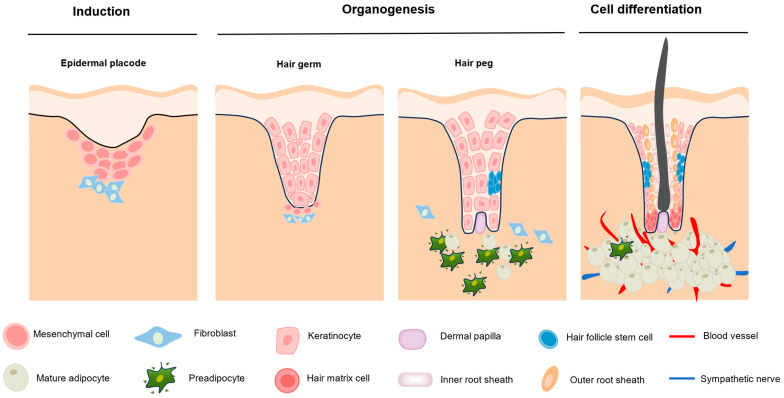
Hair follicle morphogenesis.

**Figure 2 biomolecules-15-01335-f002:**
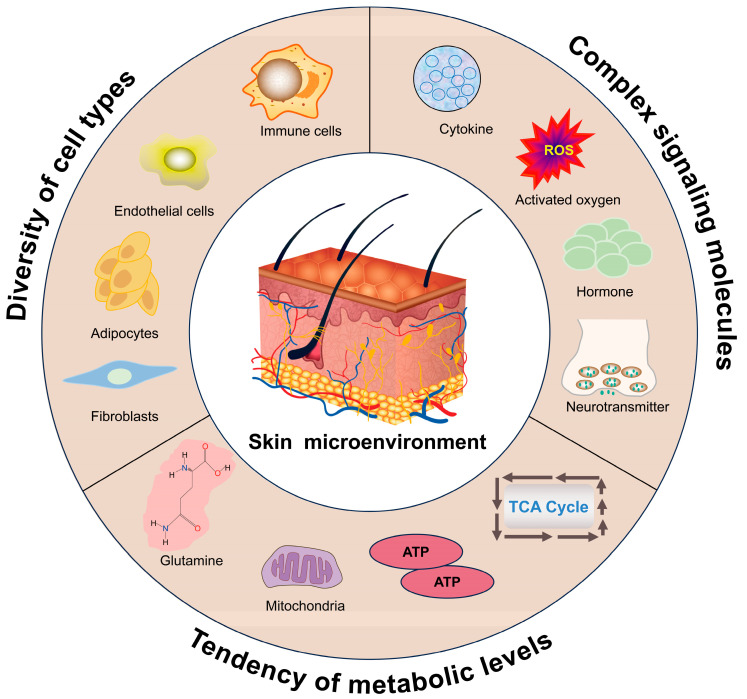
Potential skin microenvironment factors that influence hair follicle development.

**Figure 3 biomolecules-15-01335-f003:**
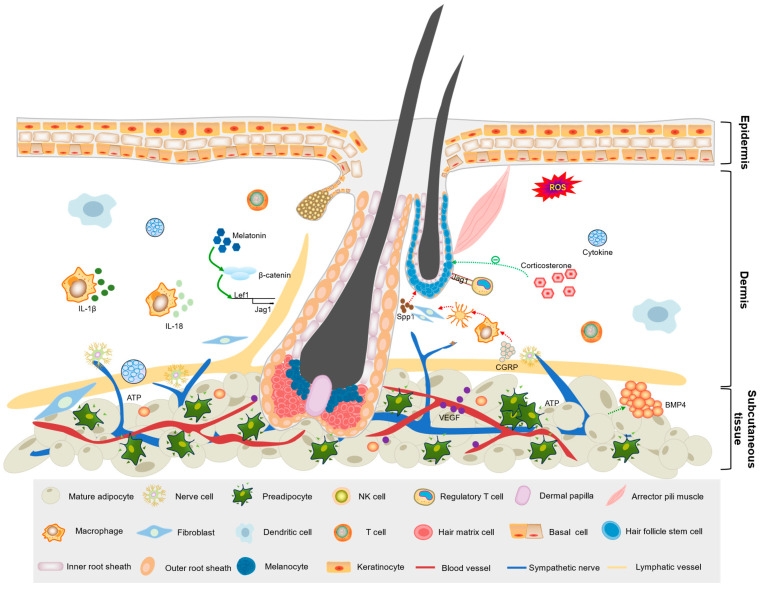
Skin follicle microenvironment.

## Data Availability

No new data were created or analyzed in this study. Data sharing is not applicable to this article.

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
