# Peer review of "The Skin Microenvironment: A Dynamic Regulator of Hair Follicle Development, Cycling and Disease"

_biomolecules, 2025, doi:10.3390/biom15091335_

Round 1

Reviewer 1 Report

Comments and Suggestions for Authors

In the present article the authors reviewed literature on the effect of the skin microenvironment in hair follicle (patho)physiology. Microenvironment has been studied extensively in cancer. However, the role of microenvironment is known to be important in many tissues.

Hair follicle morphogenesis and cycling rely on maintaining homeostasis in the skin microenvironment. An imbalance in homeostasis can result in cycle stagnation or hair loss.

Most of the studies on hair follicle physiology have focused on intrinsic signals that regulate the anagen, catagen and telogen phases of hair growth and have left untouched the role of microenvironment. The authors focus on how cell types, signaling molecules, and metabolic alterations in the skin microenvironment regulate hair follicle cells.

The microenvironment is complex and it involves multiple cell types (e.g., immune cells, fibroblasts, adipocytes, and vascular endothelial cells) and signaling molecules (e.g., hormones, neurotransmitters, cytokines, and reactive oxygen species) secreted by the cells or tissues. The author describe in more detail the contribution of each cell type on hair follicle physiology (immune cells connection with stress-induced alopecia and alopecia areata is given).

The review is interesting and I only have the following minor comments.

  1. Lines 226-227 “These changes result in aberrant HFSC activation and abnormal hair cycling” Could you please describe in more detail? Do they show alopecia?
  2. Line 252 on the role of vascular cells. Are there any data on patients receiving anti-VEGF therapy e.g. avastin and hair loss?
  3. Lines 328-330 The authors state that depletion on stem cell reservoir causes hair graying. Do keratinocytes stem cells synthesize melanin? Or melanocytes in the hair follicle that is transported to keratinocytes?
  4. In the paragraph with vitamins the authors should describe the role of para-aminobenzoic acid in hair recoloring initially observed in 1950 (Zarafonetis CJD (1950) Darkening of gray hair during para-amino-benzoic acid therapy. J Invest Dermatol 15: 399-401), but now under investigation.

Author Response

We sincerely thank the editor and all reviewers for their valuable feedback that we have used to improve the quality of our manuscript. The reviewer comments are laid out below in italicized font and specific concerns have been numbered. Our response is given in normal font and changes/additions to the manuscript are given in the green and yellow text (Reviewer 1 is highlighted in yellow text, Reviewer 2 is highlighted in green text.).

Comments 1: Lines 226-227 “These changes result in aberrant HFSC activation and abnormal hair cycling” Could you please describe in more detail? Do they show alopecia?

Response 1: We thank the reviewer for this insightful comment. The reviewer is correct to ask for more detail regarding the phenotypic outcome. Based on relevant literature in the field, the administration of PPARγ antagonists like BADGE or GW9662 prevents the normal progression of the hair cycle. Specifically, while control mice enter the anagen phase robustly, treated mice fail to activate their hair follicles and remain arrested in the telogen (resting) phase.

We have revised the manuscript to provide a more precise description of these effects, as detailed below. The changes can be found on lines 225-230 of the revised manuscript.

Comments 2: Line 252 on the role of vascular cells. Are there any data on patients receiving anti-VEGF therapy e.g. avastin and hair loss?

Response 2: We thank the reviewer for raising this critical point regarding the clinical relevance of VEGF in hair biology. Indeed, there is substantial evidence from clinical studies directly linking anti-VEGF therapy to hair loss, which strongly supports the central role of VEGF in maintaining hair growth that we propose in our manuscript.

As the reviewer astutely points out, drugs like bevacizumab (Avastin) are associated with alopecia as a frequent adverse effect. A meta-analysis cited by “Adverse Hair Reactions to New Targeted Therapies for Cancer” reported that approximately 10% (range 3.3%-26.8%) of patients treated with bevacizumab experience alopecia. This effect is not isolated to bevacizumab but is observed across a class of anti-angiogenic drugs, with sorafenib exhibiting the highest incidence (25.5%-29% of patients). This clinical data provides a compelling "reverse" narrative: if inhibiting VEGF leads to hair loss, then promoting VEGF should support hair growth. This narrative is directly confirmed by the findings

Independent research (Article Title: The Role of the VEGF, KGF, EGF, and TGF-Β1 Growth factors in the Pathogenesis of Telogen Effluvium in Women) further underscores the importance of VEGF, identifying it as the most influential growth factor among others (KGF, EGF, TGF-β1) in the development of telogen effluvium in female patients. A multivariate linear regression model was applied to assess the combined influence of these growth factors on the risk of developing telogen effluvium. All four factors were identified as significant predictors of the condition. Among them, VEGF had the strongest effect (η² = 0.836, p < 0.05), substantially greater than those of TGF-β1 (η² = 0.255), KGF (η² = 0.197), and EGF (η² = 0.130). These findings confirm that VEGF is the most influential factor associated with the development of telogen effluvium in women.

Another study (Article Title: Effects of oral minoxidil on serum VEGF and hair regrowth in androgenetic alopecia) on AGA suggests that oral minoxidil may upregulate VEGF levels, promoting hair regrowth. This prospective study included 50 participants divided into two groups: oral minoxidil (1 mg/day; n = 25) and control (n = 25). Serum VEGF levels were measured at baseline and after 12 weeks of treatment. Hair growth parameters, including hair count, diameter, shedding, and pull test results, were systematically evaluated. Results: Baseline VEGF levels were similar between groups (p = 0.1873). After treatment, VEGF levels significantly increased in the minoxidil group (217.88 ± 22.65 pg/ml vs. 142.81 ± 23.14 pg/ml in the control group, p < 0.0001). Hair count and diameter showed significant improvement (p < 0.0001 and p = 0.0040, respectively), while shedding and pull test results decreased (p < 0.0001). A positive correlation was observed between VEGF and hair count (r = 0.9965), whereas shedding exhibited a negative correlation (r = -0.5374). Conclusion: Oral minoxidil significantly elevates VEGF levels, promotes hair growth, and reduces shedding.

Therefore, the clinical data on anti-VEGF therapies and the research on minoxidil's mechanism of action form a coherent and consistent body of evidence, validating our narrative that VEGF-mediated angiogenesis is a crucial mechanism in hair follicle cycling. The hair loss observed in patients receiving bevacizumab serves as a powerful clinical corroboration of our statement.

Based on the above, we have supplemented the relevant clinical data in Section 3.2.3. Growth Factors. The reason it was not added to Section 3.1.3. Vascular Cells, as you suggested, is that Section 3.2.3 specifically covers VEGF-related research. Therefore, we believe placing it in Section 3.2.3 aligns better with logical flow. The revised content can be found in lines 405412 of the article.

Comments 3: Lines 328-330 The authors state that depletion on stem cell reservoir causes hair graying. Do keratinocytes stem cells synthesize melanin? Or melanocytes in the hair follicle that is transported to keratinocytes?

Response 3: We sincerely thank the reviewer for this insightful comment and for the opportunity to clarify this important point. The reviewer is absolutely correct.

Keratinocyte stem cells do not synthesize melanin. Melanin production is exclusively the function of melanocyte stem cells (McSCs) and their differentiated progeny, the melanocytes, which reside in the hair follicle bulge and hair bulb, respectively. These melanocytes synthesize melanin and transport it via their dendrites to the surrounding keratinocytes, which subsequently incorporate it into the growing hair shaft.

We apologize for the imprecise language in our original manuscript. The intended meaning was that the depletion of the McSC reservoir is a primary driver of hair graying. With age or due to stress, the pool of McSC is depleted or fails to be properly activated, leading to a loss of melanocytes in the hair bulb and consequently, to the production of unpigmented hair.

We have revised the relevant section (Lines 331-335) to accurately reflect this mechanism. The text now clearly specifies the role of "melanocyte stem cells" to avoid any confusion.

Comments 4: In the paragraph with vitamins the authors should describe the role of para-aminobenzoic acid in hair recoloring initially observed in 1950 (Zarafonetis CJD (1950) Darkening of gray hair during para-amino-benzoic acid therapy. J Invest Dermatol 15: 399-401), but now under investigation.

Response 4: We sincerely thank the reviewer for this valuable comment. The reviewer is absolutely correct. We apologize for the oversight in our previous manuscript version, where we inaccurately cited the pioneering work on the role of para-aminobenzoic acid in hair recoloration.

Upon careful re-examination of the literature, we confirm that the study by Zarafonetis et al. (1950) mentioned by the reviewer is indeed the first and crucial early clinical observation report of this phenomenon and should have been cited.

Following your comment, the incorrect reference has been replaced with the correct citation of Zarafonetis (1950) in the text, and the corresponding entry has been added to the reference list.

The reviewer's meticulous reading has significantly improved the accuracy of our citations and the overall scholarly quality of our manuscript. We are truly grateful for this insightful correction.

We believe the manuscript has been significantly improved following modifications based on the editors' and reviewers' comments. Please do not hesitate to contact us if any further information or clarification is needed.

Thank you for your time and consideration.

The revised manuscript has been uploaded to the submission system.

Thank you again for your guidance. We look forward to the next steps in the process.

Best regards,

Dongjun Liu
Inner Mongolia University
2025.09.08

Reviewer 2 Report

Comments and Suggestions for Authors

This study presents a novel approach to “(The Skin Microenvironment: A Dynamic Regulator of Hair Follicle Development, Cycling and Disease.” The author has justified their study in multiple approaches with in-depth supportive literature. This study could be beneficial for many researchers. However, further discussion on a practical application basis could enhance the strength of your paper:

Major comments

Q1. The author tries to provide a broad overview of the skin microenvironment in hair follicle biology. However, the novelty is somehow not expressed well. Please explain the novelty compared to already published papers.

Q2. The author tries to provide more details by incorporating morphogenesis, immune cells, adipocytes, vascular cells, hormones, vitamins, and metabolic factors. However, the narrative occasionally becomes fragmented. Examples of metabolic reprogramming (Section 3.3) appear disconnected from the earlier cell-based discussion. The author should try to make a connection with proper flow.

Q3. Much of the text reads as a catalog of findings (e.g., long descriptions of vitamin and hormone effects) without critical comparison, controversy discussion, or evaluation of study limitations. The review would benefit from highlighting contradictions in the literature (e.g., estradiol effects in different species, pro- vs. anti-inflammatory roles of macrophages) and pointing out unresolved mechanistic gaps.

Q4. Figures 1–3 are schematic but lack originality. They resemble textbook diagrams. A more integrative graphical model showing cell–signaling–metabolism interplay in the microenvironment would strengthen the review. Table 1 is very dense; condensing or reorganizing it thematically (immune-derived vs endocrine vs neuronal signals) would improve readability.

Q5. In the therapeutic section, the interventions section (microneedling, PRP, low-energy laser, nanoparticles) is informative but too brief and superficial. It reads more like a list than a critical synthesis. Mechanistic links back to the microenvironmental regulation (e.g., how microneedling alters immune infiltration or angiogenesis) should be emphasized.

Q6. The final section (challenge and prospects) is conceptually strong but overly general. For example, the statement that “hair follicle biology has potential as a model for stem cell plasticity” is interesting but requires concrete examples. The authors should propose testable hypotheses or highlight emerging technologies (e.g., spatial transcriptomics, organoid systems) with specific references.

Minor comments

Q7. The abstract is lengthy and descriptive; it should be made more concise with clearer emphasis on key messages and clinical relevance.

Q8. The keywords are too generic. Adding terms such as “immune privilege,” “angiogenesis,” or “organoids” would improve searchability.

Q9. There are occasional grammatical issues (e.g., “hair follicle is the key determinant of HFSC behavior” could be “the skin microenvironment is the key determinant…”). Careful language editing is advised.

Q10. Several references are outdated (e.g., classic studies from the 1950s are cited without linking to the modern context). Including more 2023–2024 high-impact studies will increase relevance.

Comments on the Quality of English Language

There are some grammatical errors, and author need to rewrite using proper scientific terminology.

Author Response

We sincerely thank the editor and all reviewers for their valuable feedback that we have used to improve the quality of our manuscript. The reviewer comments are laid out below in italicized font and specific concerns have been numbered. Our response is given in normal font and changes/additions to the manuscript are given in the green and yellow text (Reviewer 1 is highlighted in yellow text, Reviewer 2 is highlighted in green text).

Q1. The author tries to provide a broad overview of the skin microenvironment in hair follicle biology. However, the novelty is somehow not expressed well. Please explain the novelty compared to already published papers.

Response 1: We sincerely thank the reviewer for raising this important point regarding the novelty of our work. We appreciate the opportunity to clarify how our review distinguishes itself from existing publications in the field. Our review integrates current understanding of the skin microenvironment in a structured manner that emphasizes functional hierarchy and mechanistic depth. Existing reviews often focus on isolated aspects of hair follicle regulation, such as immune cells or signaling pathways. In contrast, we provide a synthesized view organized around three core layers of regulation: cellular components, signaling networks, and metabolic reprogramming. Additionally, we also consolidate emerging therapeutic strategies that target the microenvironment (from metabolic modulation to neural intervention) offering a cohesive framework for translating mechanistic insights into clinical applications.

Q2. The author tries to provide more details by incorporating morphogenesis, immune cells, adipocytes, vascular cells, hormones, vitamins, and metabolic factors. However, the narrative occasionally becomes fragmented. Examples of metabolic reprogramming (Section 3.3) appear disconnected from the earlier cell-based discussion. The author should try to make a connection with proper flow.

Response 2: We sincerely thank the reviewer for this insightful comment. We agree that enhancing the narrative flow between the sections is crucial for improving the coherence of the review.

In direct response to this suggestion, we have now added a dedicated transition paragraph at the beginning of Section 3.3 (Metabolic Reprogramming) to explicitly link it to the preceding discussions on cells and signaling molecules. The added text states: “The coordinated interplay between cellular constituents and signaling molecules within the skin microenvironment, as detailed above, necessitates a downstream mechanistic executor. This role is fulfilled by metabolic reprogramming, wherein dynamic shifts in cellular metabolism directly empower the biological behaviors that govern hair follicle fate.” In addition, we have further enhanced the coherence between different sections of the content.

To further aid the reviewer’s understanding, we would like to briefly outline the overall logic of our review: This review aims to provide a comprehensive understanding of the skin microenvironment's role in hair follicle development: This review introduces the origins and cyclical dynamics of hair follicle development by first reviewing its fundamental biology and then categorizing the key factors within the skin microenvironment that influence it into three primary categories: diverse cell types and tissues, complex signaling molecules, and metabolic influences. We also summarize current hair loss treatment strategies that target modifications of the skin microenvironment. Finally, we outline current research challenges and future directions.

We hope our new revisions and explanations will be of some assistance to the reviewers.

Q3. Much of the text reads as a catalog of findings (e.g., long descriptions of vitamin and hormone effects) without critical comparison, controversy discussion, or evaluation of study limitations. The review would benefit from highlighting contradictions in the literature (e.g., estradiol effects in different species, pro- vs. anti-inflammatory roles of macrophages) and pointing out unresolved mechanistic gaps.

Response 3: We sincerely appreciate the reviewer's comments and acknowledge certain limitations in this paper. As requested, we have modified the relevant sections to include additional insights on the contradictions in existing research (see lines 307-315 for specific revisions). For instance, regarding the differential effects of estradiol across species, we propose that these variations may stem from differences in estrogen receptor expression, metabolic conversion, or interactions with local growth factors. The dual pro-inflammatory and anti-inflammatory roles of macrophages, meanwhile, likely correlate closely with their spatial localization and polarization state.

We believe the manuscript is significantly improved. Thank you for this helpful suggestion.

Q4. Figures 1–3 are schematic but lack originality. They resemble textbook diagrams. A more integrative graphical model showing cell–signaling–metabolism interplay in the microenvironment would strengthen the review. Table 1 is very dense; condensing or reorganizing it thematically (immune-derived vs endocrine vs neuronal signals) would improve readability.

Response 4: We sincerely thank the reviewer for taking the time to provide these insightful comments regarding our figures and table.

As designed, each figure serves a specific role in building a comprehensive understanding of the hair follicle microenvironment: Figure 1 illustrates the collaborative actions of multiple cell types during embryonic hair follicle development. Figure 2 provides a systematic categorization of regulatory factors from three major sources within the skin microenvironment, serving as the organizational foundation for our subsequent review. Figure 3 integrates cellular and signaling information from the previous figures to visualize spatial distribution and structural details within the hair follicle, enhancing reader comprehension. We acknowledge that this approach may have resulted in a more classical style.

The more comprehensive graphical model you mentioned illustrating cell signaling-metabolism interactions has provided significant inspiration for our next review article. We would greatly appreciate your valuable input at that time.

Regarding the other issue you raised, our perspective is as follows: Many signaling molecules derive from multiple sources (e.g., a cytokine can be produced by both immune and non-immune cells under different conditions), making a strict "immune vs. endocrine vs. neuronal" categorization potentially misleading. However, the issue you raised is actually crucial for enhancing the readability of our articles, which means we must set higher standards for future writing and formatting. Thank you again for your suggestion.

Q5. In the therapeutic section, the interventions section (microneedling, PRP, low-energy laser, nanoparticles) is informative but too brief and superficial. It reads more like a list than a critical synthesis. Mechanistic links back to the microenvironmental regulation (e.g., how microneedling alters immune infiltration or angiogenesis) should be emphasized.

Response 5: We appreciate the reviewers' valuable suggestions. We have modified this section to further explain how microneedling modulates the skin microenvironment to promote hair regeneration. Please refer to lines 530-539 for the specific content.

Q6. The final section (challenge and prospects) is conceptually strong but overly general. For example, the statement that “hair follicle biology has potential as a model for stem cell plasticity” is interesting but requires concrete examples. The authors should propose testable hypotheses or highlight emerging technologies (e.g., spatial transcriptomics, organoid systems) with specific references.

Response 6: We thank the reviewer for this insightful and constructive comment. We agree that the "Challenges and Prospects" section would benefit from greater specificity and actionable guidance.

Based on this, we have rewritten and revised this section, incorporated specific experimental approaches to enrich our content, and updated the most recent references. We believe the revised version comprehensively reflects the existing challenges in the field of hair follicle biology and the scientific progress achieved through emerging technologies. The modifications can be found in the "Challenges and Prospects" chapter.

Minor comments

Q7. The abstract is lengthy and descriptive; it should be made more concise with clearer emphasis on key messages and clinical relevance.

Response 7: We thank the reviewer for this helpful suggestion. we have revised the abstract by removing unnecessary content to make it more concise and focused. We believe the updated version better conveys the core messages of our review.

Q8. The keywords are too generic. Adding terms such as “immune privilege,” “angiogenesis,” or “organoids” would improve searchability.

Response 8: We sincerely thank the reviewer for this excellent suggestion. We agree that our previous keywords were too generic and have revised them accordingly to significantly improve the article's searchability and better reflect its specific focus.

As suggested, we have added the highly relevant terms “immune privilege”. Furthermore, to provide even greater precision, we have also incorporated “hair follicle miniaturization” and “intercellular communication” (which encapsulates the core theme of microenvironmental interactions). The outdated, overly broad terms have been removed. We believe the new keyword list is now much more specific and representative of the manuscript's content.

These changes have been made in the revised manuscript.

Q9. There are occasional grammatical issues (e.g., “hair follicle is the key determinant of HFSC behavior” could be “the skin microenvironment is the key determinant…”). Careful language editing is advised.

Response 9: We thank the reviewer for this helpful feedback. We have carefully edited the manuscript for language and grammar. Specifically, the sentence highlighted by the reviewer has been revised to: "These results suggest that the skin microenvironment is the key determinant of HFSC behavior in terms of self-renewal and differentiation." We believe this revision significantly improves the clarity and accuracy of the statement.

Q10. Several references are outdated (e.g., classic studies from the 1950s are cited without linking to the modern context). Including more 2023–2024 high-impact studies will increase relevance.

Response 10: We sincerely thank the reviewer for raising this critical point. We completely agree that integrating the most recent high-impact literature is essential to establishing the modern context and relevance of our review.

In response to this comment, we have updated the literature review and incorporated selected key recent publications to enhance the timeliness and academic foundation of this review. These additions significantly strengthen the manuscript's contemporary relevance.

Regarding the citation of classic studies (e.g., from the 1950s), we have retained a select few that represent the seminal, foundational discoveries in the field (such as the first observation of PABA-induced hair darkening). Our intention is to provide appropriate historical credit to these pioneering works, which established the core principles that modern research continues to build upon. However, in each instance where a classic paper is cited, we have now ensured that it is directly discussed alongside and supported by contemporary evidence. This approach creates a more compelling narrative that connects the origin of ideas to their current development.

We believe that the revised manuscript now strikes an optimal balance between acknowledging the foundational literature and highlighting the cutting-edge advances that define the field's current trajectory.

Comments on the Quality of English Language

There are some grammatical errors, and author need to rewrite using proper scientific terminology.

Response: We sincerely thank the reviewer for highlighting this issue. We have carefully reviewed the manuscript and made further language refinements to improve grammatical accuracy and ensure precise use of scientific terminology throughout. We believe these revisions have significantly enhanced the clarity and professionalism of the writing.

We believe the manuscript has been significantly improved following modifications based on the editors' and reviewers' comments. Please do not hesitate to contact us if any further information or clarification is needed.

Thank you for your time and consideration.

The revised manuscript has been uploaded to the submission system.

Thank you again for your guidance. We look forward to the next steps in the process.

Best regards,

Dongjun Liu
Inner Mongolia University
2025.09.08

Round 2

Reviewer 2 Report

Comments and Suggestions for Authors

Thank you for the revision as per suggestion, I agree and accept the paper in present form.